# DREAM On, DREAM Off: A Review of the Estrogen Paradox in Luminal A Breast Cancers

**DOI:** 10.3390/biomedicines12061300

**Published:** 2024-06-12

**Authors:** Judith C. Hugh, Lacey S. J. Haddon, John Maringa Githaka

**Affiliations:** 1Department of Laboratory Medicine and Pathology, University of Alberta, 116 St & 85 Ave, Edmonton, AB T6G 2R3, Canada; 2Department of Chemistry, University of Alberta, 116 St & 85 Ave, Edmonton, AB T6G 2R3, Canada; lhaddon@ualberta.ca; 3Department of Biochemistry, University of Alberta, 116 St & 85 Ave, Edmonton, AB T6G 2R3, Canada; maringa@ualberta.ca

**Keywords:** breast cancer, ER positive, estrogen, DREAM, hormone therapy

## Abstract

It is generally assumed that all estrogen-receptor-positive (ER+) breast cancers proliferate in response to estrogen and, therefore, examples of the estrogen-induced regression of ER+ cancers are paradoxical. This review re-examines the estrogen regression paradox for the Luminal A subtype of ER+ breast cancers. The proliferative response to estrogen is shown to depend on the level of ER. Mechanistically, a window of opportunity study of pre-operative estradiol suggested that with higher levels of ER, estradiol could activate the DREAM-MMB (Dimerization partner, Retinoblastoma-like proteins, E2F4, and MuvB–MYB-MuvB) pathway to decrease proliferation. The response of breast epithelium and the incidence of breast cancers during hormonal variations that occur during the menstrual cycle and at the menopausal transition, respectively, suggest that a single hormone, either estrogen, progesterone or androgen, could activate the DREAM pathway, leading to reversible cell cycle arrest. Conversely, the presence of two hormones could switch the DREAM-MMB complex to a pro-proliferative pathway. Using publicly available data, we examine the gene expression changes after aromatase inhibitors and ICI 182,780 to provide support for the hypothesis. This review suggests that it might be possible to integrate all current hormonal therapies for Luminal A tumors within a single theoretical schema.

## 1. Introduction

Since Haddow first demonstrated the estrogen-induced regression of breast cancer [1], numerous other publications have reported similar findings (reviewed in [2]). In 1960, Kautz and The Council on Drugs reported on 944 post-menopausal patients with breast cancer treated with estrogens or androgens, finding that the percentage of women showing estrogen-induced regression increased from 12 to 30% if treatment began five years or more after menopause [3]. More recently, the Women’s Health Initiative (WHI) trial investigated estrogen or estrogen plus progesterone hormone replacement therapy (HRT) in women with a median age at enrollment of 63.2 years (range 50–79 years), thirteen years after the average age of menopause. While estrogen combined with progesterone HRT was associated with an increase in breast cancer, estrogen-only HRT was associated with a statistically significant decreased incidence of ER+ breast cancer [4]. 

The estrogen-induced regression of post-menopausal ER+ breast cancer is a paradoxical finding since ER+ breast cancers also respond to estrogen deprivation whether induced through oophorectomy, selective estrogen modulators, aromatase inhibitors or ER downregulation [5]. Two specific questions encapsulate this paradox: 1. How can estrogen both promote and restrict the growth of post-menopausal ER+ breast cancers? 2. Why does the growth-inhibiting effect of estrogen require a five-year gap of estrogen exposure? In this review, we explore the current explanation of the estrogen paradox, specifically one that is applicable to the Luminal A subtype that accounts for the majority of post-menopausal ER+ breast cancers. Based on our own work and the literature on the response of the breast to hormones, we propose a hypothesis that the cellular response to estrogen is dependent on the level of ER expression and the hormonal microenvironment.

## 2. Estrogen-Induced Regression in ER+ Cancers

### 2.1. Estrogen-Induced Apoptosis in Luminal B Cancers

The two questions regarding the estrogen paradox were investigated in a series of elegant in vitro experiments by Dr. V.C. Jordan and others in his laboratory who have used ER+ breast cancer cell lines conditioned by long-term estrogen depletion to simulate the five-year post-menopausal gap (reviewed in [6]). They showed that the response to estrogen alternates between growth induction and suppression. Contrary to the usual growth-promoting effect of estrogen on ER+ cell lines in vitro, after long-term estrogen depletion, subsequent estrogen exposure became growth-inhibitory with anti-tumor effects in these cells. However, continuous estrogen exposure was then followed by resistance and eventual regrowth during estrogen treatment. The initial growth-inhibitory effect of estrogen was found to involve at least three apoptotic pathways including the unfolded protein response, the activation of the extrinsic death receptor pathway and the intrinsic mitochondrial pathway. Estrogen-induced growth inhibition was recently shown to be dependent on increased ER expression which occurs during periods of estrogen depletion. Similar findings occur in primary tumors with *ESR1* amplification [7]. Continuous estrogen exposure decreases ER levels, and this correlates with resistance to the growth-inhibitory effects of estrogen. This cycling of the response to estrogen between growth induction and suppression mediated by intermittent estrogen exposure and the resulting change in the level of ER has led to the suggestion that “pre-emptively switching between estrogen and estrogen deprivation therapies before resistant tumors emerge is an effective strategy for long-term control of tumor burden” [7]. This laboratory advancement is driving the development of therapeutics that can condition or assist in estrogen-induced apoptosis [6] and has served as the basis for at least one large clinical trial, The Study Of Letrozole Extension (SOLE), a phase III randomized clinical trial of continuous vs. intermittent letrozole in post-menopausal women.

These findings are, however, complicated by the fact that there are two variants of ER+ breast cancer, Luminal A and Luminal B [8]. Subsequent studies have shown that Luminal A tumors have higher ER expression; are low-grade, low-proliferative tumors with minimal genetic abnormalities; respond well to current therapeutic regimens and occur almost exclusively in post-menopausal women [5]. The WHI trial showing a reduction in breast cancer was large enough to reveal the characteristics of the cancers that were in deficit with estrogen-only HRT. These were low-grade (grade 1 or 2), ductal, localized, ER+, HER2 negative with or without progesterone receptor positivity [4]. In general, this description is typical of Luminal A breast cancers. 

ER+ breast cancers are difficult to establish in vitro, particularly Luminal A tumors. Four in-depth surveys of existing breast cancer cell lines concluded that none of the available ER+ breast cancer cell lines were Luminal A and were most likely Luminal B [9,10,11,12]. These include the cell lines used in the long-term estrogen deprivation studies. In addition, the only two estrogen-suppressed patient-derived xenograft models reported to date, WHIM16 [13] and GS3 [14], are both luminal B cell lines [15,16], suggesting that the apoptotic pathway triggered by estrogen has been elucidated exclusively in luminal B cells. The story might therefore be different in Luminal A cells.

### 2.2. Evidence for a Non-Apoptotic Pathway of Estrogen-Induced Regression

Clinically, Luminal A breast cancers respond well to hormonal treatment with a response pattern that does not fit with an apoptotic or cytotoxic model. Current endocrine therapies promote a cytostatic effect with a reduction in tumor size, requiring months of treatment and an optimal duration of therapy of 5 to 10 years [17]. 

There are numerous reports showing that transfection of the ER gene (*ESR1*) into previously ER-naïve cells, including human mammary epithelial cells, breast and cervical cancer cells, Chinese Hamster ovary cells, rodent fibroblasts and a human osteosarcoma cell line, show an immediate estrogen-induced regression that is associated with a slower time course than is typical for an apoptotic agent [18] (Zajchowski et al., and references therein). Two other reports [19,20] found that *ESR1*, when transfected into MCF-7 cells, was associated with estrogen-induced cell cycle block. Accordingly, we transfected MCF-7 cells with an inducible *ESR1* gene under a doxycycline promoter [21] and titrated the doxycycline dose to achieve a 20-fold increase in ER protein expression by densitometry, equivalent to an Allred score of 8/8 by standard diagnostic immunohistochemistry [22]. Mock transfected MCF-7 cells retained their characteristic Luminal B low level of endogenous ER expression, previously quantitated at 63 fmol/mg protein [23], equivalent to an Allred score of 3/8. 

With the increased levels of ER, the response to estrogen became anti-proliferative with G1/S and G2/M cell cycle block and no evidence of apoptosis. Like Zhao et al. before us [19], we found that the cell cycle block was associated with an increase in p21. Chromatin immunoprecipitation using an anti-ER Ab, followed by whole-genome sequencing, showed the increase in p21 was most likely mediated through direct ER binding to the p21 gene (*CDKN1A*) in a novel intragenic region composed of two half estrogen response elements (EREs) and an AP1 site. ChIP-String analysis confirmed that ER binding in this region only occurs in cells with high *ESR1* in the presence of estrogen. 

This work demonstrated that a given hormone is not strictly paired to a single proliferation outcome and that estrogen could elicit diametrically opposite cell cycle responses (proliferation or cell block) depending on the amount of ER present. This conclusion is identical to that recently published by Traphagen et al. [7]. Those authors were able to show that high levels of endogenous ER and transcriptional activity were necessary for the growth-inhibitory effects of estrogen. With prolonged estrogen exposure, the ER levels declined, and this decrease in ER was associated with resistance to the growth-inhibitory effects of estrogen [7]. It raised the possibility that Luminal A breast cancers, which have higher amounts of ER, might have a p21-mediated, non-proliferative response to estrogen. In the following sections, we present our investigation of this hypothesis with supporting literature to answer the two key questions regarding the estrogen paradox when applied to Luminal A tumors.

### 2.3. How Can Estrogen Promote and Restrict the Growth of Luminal A Cancers? DREAM

Luminal A tumors are thought to closely resemble normal breast epithelial cells, with few, if any, genetic abnormalities [5]. This is significant because the response to estrogen in normal breast epithelium is opposite to that in uterine epithelium [24,25,26]. Histologic studies of samples taken concurrently from the breast and the endometrial lining of the uterus during the menstrual cycle have shown that during the pre-ovulatory or follicular phase when estrogen alone is present, the endometrial epithelial cells proliferate but the breast epithelium does not. In the post-ovulatory or luteal phase of the cycle when estrogen and progesterone are both present, the endometrial epithelial cells stop proliferating, while the breast epithelia start proliferating. This is consistent with the WHI trials of HRT in which combined estrogen and progesterone protected against endometrial cancer but was associated with an increase in breast cancer [4]. We decided to test the hypothesis that Luminal A tumors may arise after menopause because of the lack of estrogen and regress when the estrogen is supplied either as HRT or as a therapeutic. 

We used a Window of Opportunity Study (PRESTO–PRe-operative ESTradiOl Window of Opportunity Study in Post-Menopausal Women with Newly Diagnosed ER Positive Breast Cancer) to test the hypothesis that estrogen could induce a decrease in Ki67 or Risk of Recurrence Score (ROR) (BC360™, NanoString Technologies Inc., Seattle, WA, USA) [27]. Nineteen women with newly diagnosed ER+, low-grade breast cancer who were post-menopausal with at least 5 years from their last estrogen exposure were given 6mg/day of estradiol [28] for 7 to 14 days in the interval between diagnosis and surgery. 

In total, 13 of 19 (68%, *p* = 0.025) patients showed a decrease in Ki67, while 8/13 (62%, *p* = 0.07) patients showed a decrease in ROR score with no histologic or gene expression evidence [29] of apoptosis. Using ROR scores instead of Ki67 to define estrogen response [30,31,32,33,34], we defined the three patients with the greatest reduction in ROR as “responders” and compared their gene expression profiles (GEPs) to three “non-responder” patients who had the least change in ROR between biopsy and surgical specimens. 

We derived a GEP signature using a non-parametric random forest approach [35] that separated the responders from non-responders on the post-estrogen treatment surgical specimens (PRESTO-45^surg^, [25]). The algorithm selected genes based on their ability to predict the response status of the sample independently of their level of significance in classical parametric statistical analysis. Surprisingly, even though p21 was significantly different between responders and non-responders, it was not part of the predictive signature. The regulatory elements of two-thirds of the PRESTO-45^surg^ genes were most consistent with FOXM1 and E2F4, crucial components of the DREAM-MMB (**D**imerization partner, **R**etinoblastoma-like proteins, **E**2F4, **a**nd **M**uvB–**M**YB-**M**uv**B**) system of cell cycle control that is responsible for reversible cell cycle arrest or quiescence [36]. In 2019, ref. [37], a rigorous global analysis using knockout cells, DREAM-ChIP verification and consistency across cell types and species, defined 268 genes that were repressed through the p53-DREAM pathway. Greater than 85% of these genes were involved in cell cycle regulation. In total, 30 of these 268 genes are contained within the PRESTO-45^surg^ and were downregulated with estrogen. A total of 12 of the 268 genes were not annotated in our database. Of the remaining 226 DREAM genes, 200 (89%) were also downregulated in the responders. This near uniform suppression of the DREAM genes [37] is consistent with estrogen-induced activation of the DREAM complex to mediate cell cycle block. 

Notably, DREAM acts in concert and is partially redundant with the RB/E2F pathway, both of which are essential for the repression of many G1/S and G2/M cell cycle genes [37]. In addition, RB protein is positive immunohistochemically in over 90% of ER+ tumors [38], and RB mutants are resistant to the DREAM agonist effect of cyclin-dependent kinase inhibitors [39]. Therefore, we initially investigated whether 10 genes which are known to be RB repressed in a DREAM-independent manner [37] were also downregulated in our responders. We found no significant changes in these 10 genes in the responders or non-responders (see Appendix A). When we analyzed all 415 RB-E2F candidate genes [37], only 15 (3.7%) were significantly downregulated by estrogen in responders, with no genes showing significant changes in the non-responders. This suggests that the RB-E2F pathway is not a major element in the estrogen response and that the estrogen-induced gene-repression effects were mediated primarily through the DREAM pathway. 

The DREAM complex induces cell cycle arrest through binding and inhibiting E2F elements in the promoters of G1/S genes and the cell cycle genes homology region (CHR) promoter sites of G2/M genes [36]. Although the DREAM-MMB pathway is typically activated by p53 and p21 following DNA double-strand breaks due to genotoxic stress, oncogene activation or mitochondrial dysfunction, the PRESTO responders showed no evidence of changes in p53 protein (Figure 1a) or mRNA (Figure 1b). However, the responders had significantly increased mRNA of *CDKN1A* (p21) and decreased mRNA of *FOXM1* and *MYBL2*, which are the competing partners for MuvB (Figure 1b) which would shift the DREAM-MMB pathway to the DREAM-mediated repression of cell cycle proteins leading to a cell cycle block. Since both FOXM1 and MYBL2/B-MYB are known MuvB target genes, the p21-dependent formation of the DREAM complex would repress those genes, further enhancing DREAM complex formation. 

This suggests that estrogen therapy may be inducing quiescence, a stereotyped form of reversible cell cycle arrest that is generally mediated by the DREAM complex, possibly a throw-back to the normal function of estrogen in the induction of temporary cell cycle arrest during the follicular (estrogen-only) phase of the menstrual cycle. 

The DREAM pathway was recently shown to be activated by progesterone in an ovarian cancer model [40] and by supra-physiological androgens [41] in a prostate cancer model. Given the well-described pro-tumorigenic activities of estrogen and the acquisition of an invasive and migratory phenotype in the ovarian cancer model’s progesterone activation of DREAM, we examined our estrogen-treated cell lines and patients to determine if the induction of quiescence was similarly linked to pro-tumorigenic consequences. We carried out a bioinformatic analysis identical to that in Mauro [42], investigating the top regulated genes and enriched pathways after estrogen treatment. Out of the 200 genes listed in the PR-A/B inventory [42], only 4 genes overlapped between the PR-A ovarian model and either our estrogen-treated cell lines or PRESTO patients, suggesting different mechanisms are involved. While this represents unvalidated RNA-seq data, the presence of two upregulated cytokines, CXCL1 and CXCL2, in the PRESTO responders (Appendix B Figure A1a) is interesting, especially in light of the pathway analysis which showed that the PRESTO responders showed activation of the innate immune system with the upregulation of the cytokine signaling pathway, neutrophil degranulation and associated interleukin and G protein-coupled receptor signaling (Appendix B Figure A1b). The significance of this is unknown. Breast cancer cells are known to secrete cytokines that can promote invasion [43], and in particular, IL-11 has been shown to be associated with bone metastases in low-grade tumors [44]. Thus, it is possible that DREAM-mediated quiescence could be associated with the tumor dissemination of dormant cells. However, in other models, inflammatory cytokines inhibit tumor growth [45]. Notably, estrogen-only HRT in the WHI study was associated with a decreased incidence of and decreased mortality associated with breast cancer [4]. Further work is necessary to determine if DREAM-mediated quiescence is accompanied by tumor dissemination or anti-tumor effects. 

### 2.4. Why Does Estrogen Inhibition Require a 5-Year Gap? Estrogen in Context

HRT trials that initiate estrogen-only therapy close to the menopause such as the Million Women Study in the UK [46] have not found an estrogen-induced decrease in breast cancer, raising the question for the above hypothesis—if estrogen is functioning to activate the DREAM pathway, then why is there a need for the five-year gap? For Luminal B cancers, the work by Jordan’s lab has suggested that this period of estrogen depletion leads to an increase in endogenous ER, which is a prerequisite for the apoptotic or growth-inhibitory response to estrogen [6,7]. For Luminal A cancers which have consistently higher levels of ER and we presume are always growth inhibited by estrogen, the question becomes more perplexing. 

The first real mention of the 5-year gap was the 1960 Council on Drugs publication [3] which showed that the efficacy of hormone treatment in breast cancer increases dramatically five years after the menopause, which is generally defined as 50 years old. Thus, the gap may not specifically revolve around exposure to estrogen but rather to events in the menopausal transition—the five years before and after the final menstrual period (FMP) or 45 to 55 years of age. 

Interestingly, the incidence of breast cancers also shows an unusual discontinuity around the menopausal transition (Figure 2). This was first described by Clemmesen in 1948 [47], (Figure 2a) and was also noted in the 2012 surveys of age and breast cancer in both the Lancet Oncology review [48] (Figure 2b) and the SEER data review [49] (Figure 2c). In Clemmesen’s review of the incidence of breast cancer in the Danish Cancer Registry, he noted a statistically significant but “peculiar feature..is the fall in incidence for the age classes between 45 and 55”, which later became known as Clemmesen’s Hook. The 2012 Lancet Oncology review [48] showed that Clemmesen’s Hook occurred in ER+ breast cancers, while the SEER publication [49] pin-pointed Clemmesen’s Hook as affecting ER+ PR+ breast cancers (Figure 2c). If one compares the trajectory of the two ER+ breast cancers in the SEER data, the incidence of ER+PR+ breast cancers begins to increase in the mid-40s (Figure 2d, indicated by “a”), approximately 5 years before the precipitous drop in estrogen levels at 50 years and the rise in ER+PR- cancers. The incidence of ER+PR+ cancers then levels off for the five years (Figure 2d, indicated by “b”) following the final menstrual period (FMP), before resuming the steady linear increase with age exhibited by ER+PR- breast cancers. This suggests the possibility that the five-year gap may be reflecting the physiological events of the menopausal transition.

The human female menopausal transition is unique to female primates [50]. The Study of Women’s Health Across the Nation (SWAN) was a longitudinal study of women going through the menopausal transition in which 15,930 observations from 2886 women were aligned with ovarian status as defined by the World Health Organization. SWAN described a novel surge in adrenal androgens in 80% of women between the ages of 45 to 55 years [50]. A similar increase in four androgens, including total testosterone, dehydroepiandrosterone (DHEA), dehydroepiandrosterone-sulfate (DHEAS) and androstenedione, in women from 45 to 55 years was recently described in a population-based cohort study of 3291 participants in the Rotterdam Study [51]. Lasley, a principal investigator of the SWAN study, noted that this androgen surge also occurred following ovariectomy [52] and theorized that “changes in ovarian function... (perhaps a loss of inhibin B and a slight rise in FSH)... could trigger a transient increase in adrenal δ-5 steroid production that continues to and past the final menstrual period (FMP)”. 

Clinically, the menopausal transition is associated with signs and symptoms similar to estrogen deficiency and is alleviated by estrogen therapy even though the early transition period predates the major decrease in E2. Lasley suggested that these symptoms are potentially related to levels of androstene-3β, 17β-diol (Adiol), an adrenal androgen and metabolite of DHEAS, with the ability to bind both estrogen and androgen receptors (ARs) [50]. During the menopausal transition, levels of Adiol are elevated from less than 1 nM in premenopausal women to 3–4 nM, about one hundred times the average concentration of estradiol [50]. 

We suggest that the hormonal milieu at the beginning of the menopausal transition with the combination of high levels of Adiol in a woman with normal levels of estrogen promotes the proliferation of normal breast epithelial cell and their derivative luminal A breast cancers, leading to the early rise in ER+PR+ breast cancers. With the abrupt loss of estrogen at the time of the FMP and the persistence of Adiol for the five years beyond the FMP (late menopausal transition), there is a drop in ER+PR+ breast cancers (Clemmesen’s Hook). Until Adiol levels have decreased sufficiently, five years after the FMP, estrogen supplementation is associated with an increase in breast cancer. Once the Adiol surge has subsided, five years after the FMP, estrogen as a single hormone is associated with the suppression of proliferation in high-ER-expressing luminal A breast cancers. This explains the need for a five-year gap before estrogen therapy begins to suppress ER+ breast cancers. This phenomenon is restricted to ER+PR+ cancers because of the role of liganded and unliganded PR in regulating DREAM expression [40].

## 3. A Hypothesis for the Estrogen Paradox in Luminal A Cancers

### 3.1. DREAM ON, DREAM OFF

The previous section suggests that Clemmesen’s Hook around the menopausal transition exemplifies the rule that while two hormones (estrogen and Adiol) in the early menopausal transition cause proliferation and an increase in breast cancer, one hormone (Adiol in the last 5 years of the menopausal transition) causes cell cycle block and a decrease in breast cancer. Notably, a similar finding holds for progesterone. In a large survey of post-menopausal women, Trabert et al. [53] found that in women with higher levels of estrogen, increasing serum progesterone was associated with an increase in breast cancer. However, for women in the lowest quartile of circulating estrogen, increasing serum progesterone was associated with a decrease in breast cancer. This is reminiscent of the effect on breast epithelium of the menstrual cycle in pre-menopausal women. In the estrogen-only follicular phase, the cells are non-proliferative, but this switches to proliferation in the luteal phase in the presence of both progesterone and estrogen. This suggests that the specific hormone itself (whether estrogen, an androgen or progesterone) is not important but rather the presence of another steroid hormone that determines whether the response will be cell block (one hormone) or proliferation (two hormones). 

The switch between proliferation and cell cycle blockade could be mediated by the DREAM-MMB pathway, which is designed to alternate between a pro-proliferative and anti-proliferative gateway. The system revolves around the ability of the LIN54 member protein of MuvB to stabilize different mutually exclusive complexes on the CHR promoter element of late cell cycle genes [36]. When MuvB is associated with the DREAM components (Dimerization partner, hypophosphorylated p130 Retinoblastoma-like proteins, and E2F4), the complex suppresses gene transcription and induces quiescence, e.g., “DREAM ON”. When MuvB is associated with B-Myb and/or FOXM1, these replace the inhibitory DREAM complex, the “DREAM OFF” situation, and facilitate the activation of the same cell cycle genes [36]. 

The DREAM-MMB pathway is usually activated by p21 downstream of p53 [37]. However, binding sites for ER [21,54], AR [55] and PR [56] have been described upstream of the p21 transcription start site and in intragenic regions, accounting for the observation that p21 can be activated by estrogen, androgens or progesterone. Thus, the activation of p21 by any one of these hormones could engender cell cycle blockade via the DREAM complex. The addition of a second steroid hormone could oppose the action or formation of the DREAM complex in favor of the pro-proliferative MMB complexes. 

### 3.2. One Hormone vs. Two Hormones

Progesterone is a known antagonist of the estrogen transcriptome [57]. Similarly, the opposition of the estrogen response by Adiol has been well documented in the literature. Initially, it was suggested that Adiol’s reversal of an estrogen effect was due to competition for the ER [58,59]. Subsequently, it was suggested that the higher-affinity estrogen would displace Adiol from the ER and promote Adiol–AR binding, which would oppose estrogenic effects [60,61,62]. Using 5α-dihydrotestosterone (DHT), others have suggested that liganded AR could compete for estrogen response elements, thereby preventing the activation of estrogen–ER target genes at a genome-wide level [63,64]. Most recently, Hickey et al. [65] have added competition for transcription factors or squelching as a mechanism for the androgen reversal of estrogen signaling. 

Thus, the concurrent presence of any two steroid hormones could reverse the pro-DREAM effect of a single hormone by competing for the cognate receptor or response element on the chromatin, or essential transcription factors. Alternatively, or in addition, the second hormone with its receptor could stimulate the transcription of DREAM complex antagonists, e.g., the pro-proliferative transcription factor partners for MuvB such as FOXM1 and/or MYBL2 (see Figure 3). 

### 3.3. Estrogen and Anti-Estrogens

The foregoing discussion raises a third question for the estrogen paradox. The PRESTO and the WHI data suggest that it is the Luminal A patients who respond to estrogen—the same patients who currently have the best response to existing anti-estrogen treatment protocols such as tamoxifen, aromatase inhibitors (AIs) and ICI 182,780 (Fulvestrant, Faslodex). So, how does the estrogen induction of DREAM apply to anti-estrogen therapies? It is a question that dates back to the original head-to-head trial of an estrogen (diethystilbestrol, DES) vs. an anti-estrogen (tamoxifen) that validated the use of tamoxifen, showing it was equally as effective as DES with fewer side effects [66]. The 20-year follow-up showed that the estrogen was better [67], so it is possible that it is tamoxifen’s known weak estrogen agonist properties that are therapeutic. But the same could not be said of aromatase inhibitors or ICI 182,780, which decrease the amount of estrogen or the estrogen receptor, respectively. However, the hypothesis presented above is not hormone specific and instead would argue that if there was evidence of estrogen stimulation in the post-menopausal period when adrenal-derived androgens are the dominant hormone, then the removal of estrogen (or less practically, the androgen) could activate the DREAM pathway, leading to cell cycle block and quiescence. 

Circulating estrogen is increased in post-menopausal women with breast cancer compared to those without [68]. Also, intra-tumoral estrogen is 8-fold higher than serum levels in post-menopausal ER+ tumors [69], especially in ER+PR+ tumors that have immunohistochemically detected aromatase enzyme [70]. Systemic AI therapy by inhibiting the aromatase enzyme from converting androgens to estrogen is able to decrease both circulating and intra-tumoral estrogen [71]. Dr. Angela Brodie, the pioneering scientist behind AIs, and her co-authors proposed [72] that there is an ongoing balance between estrogenic and androgenic signaling at the cellular level. They suggested that the therapeutic effect of AI is to decrease the influence of estrogen, thereby “unmasking the inhibitory effect of androgens acting via the AR”. 

Using data from Hickey et al. [65], we looked at the effect of androgen–AR signaling on DREAM genes by comparing significant fold changes (*p* < 0.05) after DHT treatment of ER+ patient-derived xenografts (PDX) HCI-005, GAR15-13 and the luminal B cell line ZR-75-1 ([65], Supplementary Table S3). Unlike the DHT-treated ZR-75-1 cell line, both the DHT-stimulated PDX models downregulated DREAM-regulated genes, similar to estrogen-induced changes in PRESTO responders and the ER-transfected MCF-7 cells (ER10) (Figure 4), confirming that single hormonal stimulation with estrogen in vitro and in vivo and the androgen DHT in vitro can activate the DREAM complex (DREAM “ON”) and downregulate proliferation genes. 

We then analyzed responders from a window of opportunity study that included mRNA data pre- and post-2-week AI treatment on 121 patients from the PeriOperative Endocrine-Therapy for Individualized Care (POETIC) trial [73]. Using data from the public repository (GSE105777), we determined ROR scores for the pre- and post-treatment samples. Similar to the PRESTO study, we categorized 89 (74%) as responders who showed a decrease in ROR, whereas 32 (26%) were non-responders who showed no decrease or an increase in ROR after treatment. The non-responders showed no change in the expression of the DREAM genes and clustered with the PRESTO non-responders (Figure 4). In contrast, the AI responders had a decrease in expression of DREAM genes that clustered with the DHT-treated PDX GAR15-13 and the estrogen responders in the PRESTO trial (Figure 4), supporting the hypothesis that unopposed androgen after AI and estrogen in the PRESTO trial can activate the DREAM anti-proliferative pathway. 

This could also explain the reduction in breast cancer incidence in post-menopausal women at high risk for breast cancer given AIs [74,75,76]. The removal of estrogen resulting in an unopposed endogenous androgen milieu could decrease the ER+PR+ Luminal A breast cancers in much the same way as the five years of the late menopause transition (Clemmesen’s Hook). 

By selectively degrading ER, ICI 182,780 could similarly unmask an androgen-derived anti-proliferative effect. In addition, ICI 182,780 may have some non-overlapping effects with AI [77]. Fulvestrant, particularly the 500 mg dose, may have superior activity in ER+PR+ tumors [78] and has been shown to prolong overall survival when added to Anastrazole in post-menopausal hormone-receptor-positive metastatic breast cancer patients [79]. Early work by Carroll et al. [80] found that ICI 182,780 directly induced quiescence in MCF-7 cells by reducing cyclin D1 mRNA. The effective increase in “free” p21 resulted in a switch to p21-cyclin E-Cdk4 complexes and an increase in quiescent cells. Thus ICI 182,780 could activate DREAM through both the diminution of an estrogen influence with unmasking of an androgen anti-proliferative effect as well as by directly inducing a cyclin E/p21-mediated quiescence. 

Accordingly, we compared the change in the expression of DREAM genes in patients treated pre-operatively with 500 mg of Fulvestrant in the Neo-adjuvant Endocrine therapy for Women with Estrogen-Sensitive Tumors (NEWEST) trial [77]. As with the PRESTO and POETIC data, we calculated ROR scores and divided the patients into seven (70%) responders (post-treatment decrease in ROR) and three (30%) non-responders (no change or increase in ROR score post-treatment). Again, the Fulvestrant-treated responders clustered with the in vitro single-hormone responses to androgen (e.g., DHT-treated PDX lines HCI-005 and GAR15-13), and estrogen in high-ER-expressing cells (ER10), as well as the responders to estrogen in the PRESTO trial and AI responders from the POETIC trial, supporting a common pathway for the anti-proliferative response to hormone therapy. Notably, although there was comparable downregulation of assessable DREAM genes in both the AI (217/227, 95.6%) and ICI 182,780 (185/195, 94.9%)-treated responders, the NEWEST responders showed more effective downregulation of these genes (Figure 4). 

**Figure 4 biomedicines-12-01300-f004:**
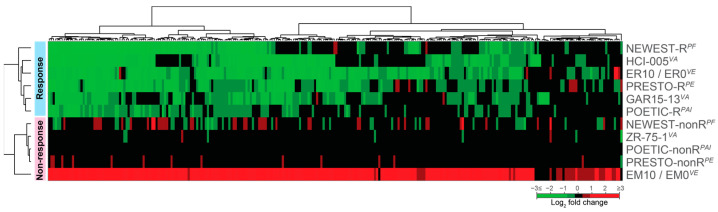
Heat map of gene expression changes in the 268 DREAM genes [37]. There is activation of the DREAM pathway with the downregulation (green) of DREAM-regulated genes with single-hormone administration in vitro of DHT in ER+ PDX HCI-005 and GAR15-13 [65] or estrogens (ER10, ER-transfected MCF-7 cells [21]), as well as ER+ breast cancers in post-menopausal patients who respond with a post-treatment decrease in ROR to estrogen (PRESTO-R [27]), aromatase inhibitors (POETIC-R [73]) or ICI 182,780 (NEWEST-R [77]). Patients who have not responded with a decrease in ROR score to either ICI 182,780 (NEWEST non-R [77]), aromatase inhibitors (POETIC non-R [73]) or estrogen (PRESTO non-R [27]) show minimal change in these genes. Even though the ZR-75-1 cell line expresses the AR and is growth-inhibited by DHT, there are no changes in DREAM genes with DHT [65]. In contrast, estrogen treatments of mock-transfected MCF-7 cells (EM10 [21]) show the upregulation of virtually all the DREAM genes. *V*—in vitro, *A*—treated with DHT (dihydrotestosterone), E—treated with estrogen, *P*—post-menopausal ER+ breast cancer patients, *F*—treated with Fulvestrant (ICI 182,780), *AI*—treated with aromatase inhibitors.

As to how estrogen treatment in the androgen-rich milieu of post-menopause activates the DREAM pathway, there is in vitro evidence ([72] (p. 7778) and [59] (p. 1852)) that higher levels of estrogen can overcome or reverse a concomitant androgenic signal. This may be analogous to DREAM activation by supra-physiological levels of androgens in a prostate cancer model [41]. This primacy of estrogen stimulation would explain the role of added (supra-physiological) estrogen in situations such as the PRESTO trial, the WHI estrogen-only arm and the relief of estrogen insufficiency symptoms during early peri-menopause when there are endogenous levels of both estrogen and androgen present. 

We also investigated the contribution of the RB-E2F pathway in these data using Principal Component Analysis (see Figure 5). The gene changes within the DREAM pathway account for the majority of variability (PC1 = 75.7%), with a comfortable separation of responders from non-responders. Conversely, even though the 415 genes in the RB-E2F pathway accounted for 39.8% variability in PC1, they grouped responders and non-responders together, with MCF-7 high-ER- and low-ER-expressing transfectants flanking this group (see Figure 5). This suggests that the estrogen response differences based on the level of ER in the luminal B MCF-7 cells are associated with greater use of the RB-E2F pathway than seen in previously untreated (presumably Luminal A) patient samples. 

Finally, the use of the DREAM pathway by current endocrine therapies would explain the clinical synergy between cyclin-dependent kinases inhibitors (CDKis) and hormone therapy [81] because of their convergence at the G1-S transition. The third-generation CDKis, palbociclib, ribociclib and abemaciclib, and p21 both function to block/inhibit the function of cyclin D-CDK4/6. This heightened inhibition of kinase function would prevent phosphorylation of the retinoblastoma family proteins (RBL1 (p107) and RBL2 (p130)) and the subsequent dissociation and activity of the E2F transcription factor. Thus, both therapeutics would accentuate DREAM-complex formation and quiescence (see Figure 6a). In a recent real world analysis, the synergy between AI and palbociclib was particularly marked in Luminal A (low-grade, PR-positive) breast cancers [82].

## 4. Conclusions

We propose that Luminal A breast cancers with higher levels of ER respond to single-hormone stimulation by estrogens, androgens or progesterone by activating the DREAM complex, inhibiting the expression of cell cycle genes and inducing quiescence (Figure 6a). Furthermore, when a physiological level of estrogen is combined with progesterone (e.g., combined HRT) or an androgen (e.g., estrogen plus Adiol in the early menopausal transition), the combination of those hormones would result in replacement of the DREAM complex by pro-proliferative MuvB-FOXM1 and MuvB-MYBL2 complexes (Figure 6b), leading to an increase in ER+ breast cancer. Thus, in this schema, each of these three hormones can be considered as pro- or anti-proliferative, depending on the presence or absence of the other hormones. Finally, we suggest that the effect of anti-estrogens is to reduce the hormonal stimulation to a single hormone through the removal of estrogen (AI and ICI 182,780) and/or the direct initiation of DREAM complexes (ICI 182,780). 

## 5. Future Directions

This review is primarily hypothesis-generating, and there is considerable work needed to prove the involvement of the DREAM pathway in hormone therapy of post-menopausal ER+ breast cancer. However, DREAM is only one possible mechanism among others to explain endocrine therapy response. Ideally, this review illustrates that the way forward can and should include a unified theory of hormone response in Luminal A cancers that encompasses all endocrine therapies, including estrogen, within the complex post-menopausal hormonal environment. 

## Figures and Tables

**Figure 1 biomedicines-12-01300-f001:**
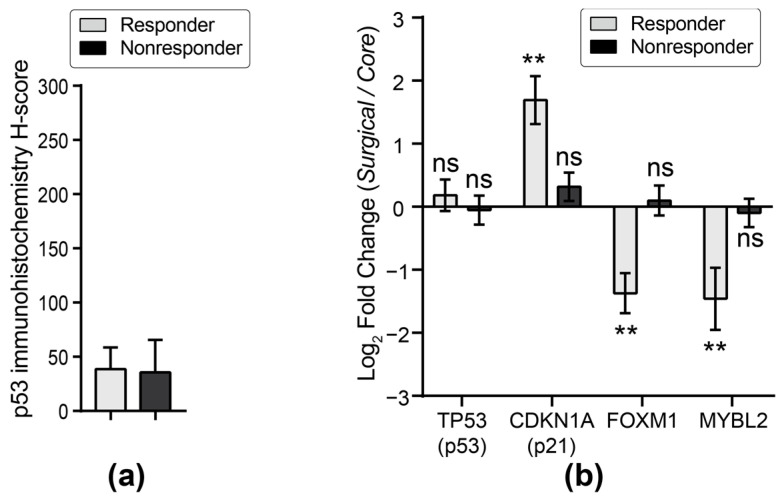
Changes in key DREAM mediators. (**a**) p53 protein expression (DO-7antibody) in the surgical specimens of responders and non-responders is similar (*p* = ns, *t*-test). (**b**) Log2 fold change (surgical over core samples) in mRNA of responders or non-responders for *TP53* (p53), *CDKN1A* (p21), *FOXM1* and *MYBL2* after estradiol. ** *p* < 0.01, ns—not significant. Reprinted from Figure 5A,B, [27] p. 6. Creative Commons License.

**Figure 2 biomedicines-12-01300-f002:**
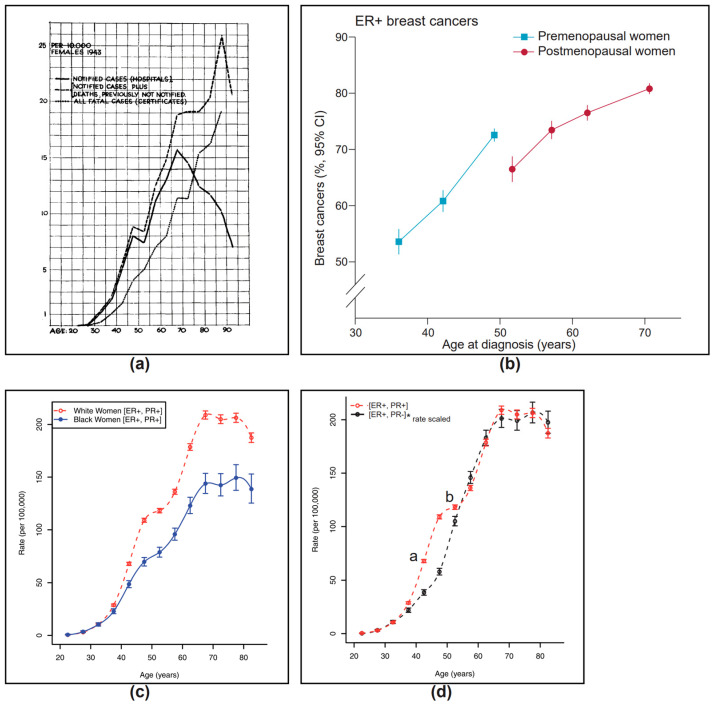
Epidemiology of breast cancer. (**a**) Hospital cases of cancer mammae according to age when first seen in hospital from the Danish Cancer Registry 1942–1944. Reprinted [38] (Figure 2) License Order Number: 5775001136853 from the Oxford University Press and the Copyright Clearance Center. (**b**) Breast cancer by tumor characteristics and by women’s age and menopausal status, estrogen receptor-positive. Reprinted [39] (Figure 6), under Creative Commons License CC BY. Modified by cropping. (**c**) Age-specific incidence rates by breast cancer phenotype in black and white women. Reprinted [40] (p. 5, Figure 3A) under Creative Commons Attribution License. Modified by cropping. (**d**) Comparison of age-specific incidence rates for ER+PR+ and ER+PR- breast cancers in white women. ER+PR+ breast cancers show an earlier rise than ER+PR- cancers during the early menopausal transition (indicated by “a”) and then level off (indicated by “b”) in the late menopausal transition before resuming the steady linear increase with age exhibited by the ER+PR- breast cancers. Reprinted [40] (Figure 3A,B) under Creative Commons Attribution License. Modified by *rate-scaling the ER+PR- curve so that the y-axis maximums are the same for both ER+PR+ and ER+PR- curves, cropping and superimposition of [40] (Figure 3A,B).

**Figure 3 biomedicines-12-01300-f003:**
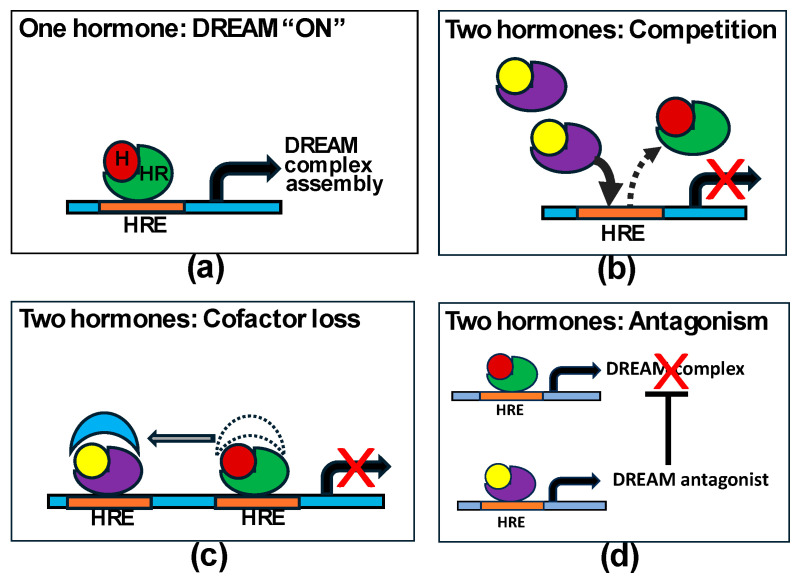
(**a**) Schematic showing one hormone (H) and its receptor (HR) binding to its hormone response element (HRE) activating the DREAM pathway. (**b**–**d**) show possible mechanisms by which two hormones could prevent DREAM activation (modified from Hickey et al. [65]).

**Figure 5 biomedicines-12-01300-f005:**
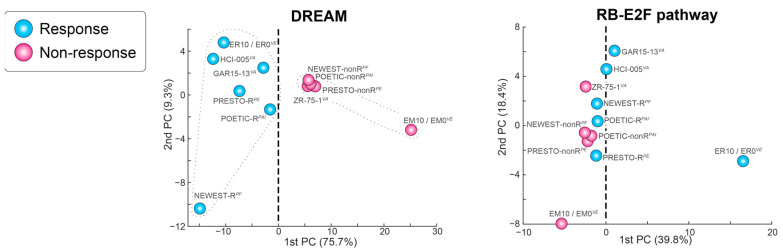
Principal Component Analysis showing the separation between responders and non-responders for the DREAM and the RB-E2F pathways.

**Figure 6 biomedicines-12-01300-f006:**
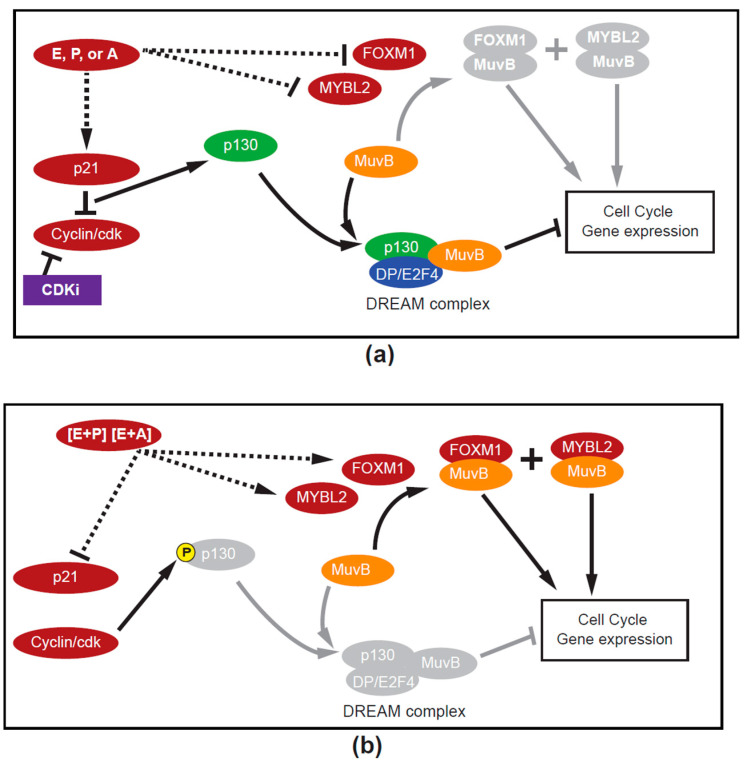
(**a**) DREAM “ON”. A hypothetical model proposing that single-hormone stimulation, either estrogen (E), progesterone (P) or androgens (A), is associated with increases in p21. The increase in p21 and its inhibition of cyclin-dependent kinase activity will lead to hypophosphorylation of the retinoblastoma family proteins, allowing them to preferentially compete for the LIN52 member of MuvB core proteins. The resulting stable DREAM repressor complex binds to the CHR element of cell cycle genes through another MuvB core protein (LIN54) and E2F binding sites via E2F4/5-DP to suppress cell cycle genes, causing reversible cell cycle arrest (quiescence) and suppression of the MuvB target genes FOXM1 and MYBL2/B-MYB. Formation of the DREAM complex would be facilitated by cyclin-dependent kinase inhibitor (CDKi) therapy. (**b**) DREAM “OFF”. A hypothetical model proposing that dual hormone stimulation (e.g., estrogen plus progesterone or estrogen plus an androgen) could (dashed line) inhibit p21 (see Figure 3). The inhibition of p21 would allow cyclin-dependent kinases to phosphorylate the retinoblastoma-like protein p130 releasing the E2F transcription factor, which would increase the pro-proliferative MuvB co-factor MYBL2. This would competitively bind the MuvB core proteins, removing them from the dimerization partner (DP/E2F4, p130) complex and activating the same cell cycle genes and increasing both FOXM1 and MYBL2/B-MYB, resulting in proliferation. Both figures reprinted and modified from [25] (p. 6, Figure 5C). Creative Commons License. Modified by shading and substitution of hormone combinations instead of estrogen as the initiating stimulus for influencing cell cycle gene expression.

## Data Availability

Data associated with this study have been deposited at the NCBI Gene Expression Omnibus (GEO) database under the accession numbers GSE139688, GSE17705, GSE2990, GSE2034, GSE105777 and GSE268001 and the ArrayExpress database under E-MTAB-887.

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
