# Peer review of "DREAM On, DREAM Off: A Review of the Estrogen Paradox in Luminal A Breast Cancers"

_biomedicines, 2024, doi:10.3390/biomedicines12061300_

Round 1
Reviewer 1 Report
Comments and Suggestions for Authors
In the manuscript submitted by Judith Hugh and colleagues, the authors review what is known about the estrogen regression paradox in ER (estrogen receptor) positive breast cancers, with a particular focus on the Luminal A subtype. They summarize data showing that cancer cells with low levels of ER respond to estrogen with increased proliferation, while proliferation of ERhigh cells is reduced. Furthermore, they describe that exposure to a single hormone - estrogen, progesterone, or androgen – represses proliferation, while exposure to a combination of these hormones has the opposite effect. They hypothesize that the different responses are caused by the formation of different transcriptional MuvB complexes – either the repressor DREAM, which shuts down cell cycle gene expression, or the activator complexes MYB-MuvB / FOXM1-MuvB, which stimulate G2/M gene expression to promote proliferation. Mechanistically, they suggest that the single hormones activate the CDKN1A promoter leading to increased expression of the CDK inhibitor p21, hypophosphorylation of pocket proteins, formation of DREAM, cell cycle gene repression, and cell cycle arrest. In contrast, binding of multiple hormones does not have these effects.
While discussing the state-of-the-art knowledge about the estrogen paradox is both interesting and important, large parts of the review focus on a paper published by Judith Hughes et al. (Heliyon, 2021, PMID: 35028452) and the doctoral thesis of the second author, Lacey Haddon (Haddon L: Increased estrogen receptor expression leads to a novel DNA binding signature which differentiates luminal 609 A and luminal B breast cancers, 2018), and the presented mechanisms are highly speculative. For example, while it has been shown that estrogen, progesterone, and androgen can activate the CDKN1A promoter, I am not aware of published data showing that a combination of these proteins has no or opposite effects on p21 expression. The data presented by the authors showing the downregulation of almost all DREAM target genes upon single hormone administration in ER+ breast cancers is convincing, however, it is unclear whether DREAM-dependent gene repression is causal for the reduced proliferation, or if proliferation is slowed down by other mechanisms, which leads to cell cycle arrest or a prolonged G1 phase accompanied by reduced expression of MuvB-regulated cell cycle genes. Overall, the manuscript appears to be more of a promotion of the author’s work and ideas, while it seems to offer less of a comprehensive review of published data.
Specific concerns:
1. The authors completely focus on DREAM and do not even mention the highly related complexes of E2F and RB proteins. Many of the target genes of DREAM and E2F/RB complexes overlap, and identical upstream mechanisms lead to their formation (CDK inhibition, etc.). The authors need to discuss what is known about the RB status in ER+ breast cancer and discuss whether RB plays or could play a role in the described mechanisms.
2. The authors should also discuss what is known about CDK inhibition using small molecules such as Palbociclib in Luminal A ER+ breast cancers since CDK inhibition also induces an accumulation of DREAM and RB complexes.
3. The figures in the manuscript are almost completely adapted from earlier publications or Lacey Haddon’s PhD thesis, and most of them I do not find helpful and unnecessary in this review article. For example, Figs. 1 and 2 show data from Lacey Haddon’s PhD thesis which are already described in detail in the text. It would be much more helpful if these findings were visualized in a sketch comparable to Fig. 6. Additionally, I think showing a sketch that compares the different mechanisms leading to regression of Luminal A vs Luminal B tumors could be helpful for the readership. Furthermore, if the authors and the editor find it necessary to show this data, Figs. 1 and 2 should be combined. Also, a representative blot should be added to Fig. 1a, along with data showing the increase in p21 levels (line 120 'data not shown'), since this increase is central to their model and much more important than the expression control shown in Fig. 1a. The title of the y-axis in Fig. 1b is unclear because it does not get explained what “E2-EtOH” means.
4. I could not find an explanation in the text for why estrogen treatment stimulates proliferation of ER-negative cells. The authors should comment on that.
5. Line 116: The authors state: "At ER levels that approach those of Luminal A cancers...". Where is it shown what amount of ER in the ER-expressing MCF-7 cells is comparable to ER found in Luminal A cancer cells? The authors should provide these data or reference published results.
6. Section 2.3 contains many experimental details that may distract the readership from the central findings, which could be explained in a few sentences.
7. The details described in lines 164-169 are already covered in the Heliyon paper and should be omitted.
8. Line 185-187: “We derived a GEP signature that separated the responders from non-responders on the post-estrogen treatment surgical specimens (PRESTO-45surg, [25]). Surprisingly, it did not contain p21 nor any commonly ER regulated genes.” Line 205-206: “However, the responders had significantly increased mRNA of CDKN1A (p21)”. I find this a little confusing – I don’t understand why the authors mention that they do not find changes in CDKN1A/p21 expression in their initial screen, but then find such changes in another assay. They should consider omitting the first part.
9. Line 192: “knockout” instead of “knock-down” cells
10. Line 196: It is unclear what “the remaining” DREAM genes mean.
11. Fig. 3b: What data is shown here? mRNA or protein?
12. Line 225-227: It would be helpful to learn more about the mechanisms leading to the cycling between growth promotion and inhibition in Luminal B cancers. Also, a reference needs to be added here.
13. The hypothetical model described in lines 354-358 could be illustrated in a figure.
14. Line 384: Knowledge of Angela Brodie's lifetime may not be essential.
15. Fig. 6: The authors should mention that both FOXM1 and MYBL2/B-MYB are also known MuvB target genes. Consequently, the p21-dependent formation of DREAM and E2F-RB complexes would repress these genes, resulting in a depletion of FOXM1 and B-MYB proteins. When CDKs are active, MYBL2 expression is stimulated by activator E2Fs, and FOXM1 itself is a target of activator MuvB complexes. Even though additional mechanisms like protein stability are likely to play a role here, the statement “the mechanism for these changes is currently unknown” (line 491, line 500) is misleading.
16. Line 494-495: Since the authors specifically explain that DREAM binds to CHR elements via LIN54, they should also mention that DREAM additionally interacts with E2F binding sites via E2F4/5-DP.
17. Line 479-486: I appreciate that the authors state that their model is hypothetical, but presenting a complex mechanism in a review article, highlighting it in the title, but then ending the article with "Even if ultimately DREAM is not shown to be involved..." does not feel right to me. I would suggest focusing the article more on the estrogen paradox and mention DREAM-dependent gene regulation as one possible mechanism among others that could provide an explanation.
Author Response
While we agree with Reviewer #1’s comment that the DREAM pathway mechanism for estrogen response in this paper is largely a reiteration of our previous publication, there is no other work currently linking DREAM to estrogen or any other treatment for breast cancer. The advance in this paper is our analysis of publicly available datasets to show that the DREAM pathway also underlies the response to aromatase inhibitors and ICI 182,780. The proposal of a single schema for carcinogenesis and treatment of Luminal A breast cancers that is consistent with the known hormonal physiology of the peri-menopause and menopausal period is unique to this report.
Please see the point-to-point reply in the attachment.

Reviewer 2 Report
Comments and Suggestions for Authors Reading this - it is a very nice review - very timely and well written - However a key concept is missing - when cells are engaging DREAM the cell cycle is inhibited, yes - BUT - these growth arrested cells now acquire a highly invasive and migratory phenotype - This is something that has been missed herein - it may not be a good idea to inhibit cell growth with cytostatic treatments that then induce EMT and metastasis of cells that are not in the cell cycle - they are linked biologies - I would ask them to add this concept to their discussion - This is clearly stated and shown in REF 35 - in a fallopian tube model system - progesterone engaged DREAM to limit cell proliferation but the cells then formed emboli and migrated and invaded into collagen much more.Author Response
Please see attachment

Round 2
Reviewer 1 Report
Comments and Suggestions for Authors
The authors have addressed all of my concerns and I support publication of the revised manuscript.